# The Beneficial Dietary Effect of Dried Olive Pulp on Some Nutritional Characteristics of Eggs Produced by Mid- and Late-Laying Hens

**DOI:** 10.3390/foods13244152

**Published:** 2024-12-21

**Authors:** Anna Dedousi, Charalampos Kotzamanidis, Georgia Dimitropoulou, Themistoklis Sfetsas, Andigoni Malousi, Virginia Giantzi, Evangelia Sossidou

**Affiliations:** 1Veterinary Research Institute, Hellenic Agricultural Organization, DIMITRA, 57001 Thessaloniki, Greece; kotzam@elgo.gr (C.K.); vgiantzi@elgo.gr (V.G.); sossidou@elgo.gr (E.S.); 2Research & Development, Quality Control and Testing Services, QLAB Private Company, 57008 Thessaloniki, Greece; g.dimitropoulou@q-lab.gr (G.D.); t.sfetsas@q-lab.gr (T.S.); 3Laboratory of Biological Chemistry, Medical School, Aristotle University, 54124 Thessaloniki, Greece; andigoni@auth.gr

**Keywords:** olive pulp, fatty acid profile, egg cholesterol, total phenolics, health lipid indices, layers, hens’ age

## Abstract

This research evaluated the impact of incorporating dried olive pulp (OP) into the feed of laying hens on the fatty acid profile, cholesterol, triglyceride, total phenolic, oleuropein and hydroxytyrosol content, and health lipid indices of eggs produced by mid- (39 weeks) and late-laying (59 weeks) birds. Over a 36-week trial, 300 eggs from 180 Isa-Brown hens, assigned to three dietary groups with different OP levels (CON, OP4 and OP6), were analyzed. OP reduced egg cholesterol, with significant effects in late-age eggs (*p* < 0.05). In mid-age hens, the OP6 eggs had higher total phenolics than the controls (*p* < 0.05) and more PUFAs than the other groups (*p* < 0.05). The concentration of total phenolics, cholesterol, n3 PUFAs and % fat increased with hen age (*p* < 0.05), while triglycerides and oleuropein decreased (*p* < 0.05). With increasing hen age, the SFAs in the OP eggs decreased (*p* < 0.05) and the MUFAs increased (*p* < 0.05). Eggs from older hens had higher nutritional value, as indicated by the lower n6/n3 PUFA ratio, lower AI and TI indices, and higher h/H ratio (*p* < 0.05). Overall, dietary OP supplementation improved the nutritional quality of eggs, suggesting potential health benefits. Our results also highlighted eggs from older hens as a valuable source of high-quality fats.

## 1. Introduction

The Western lifestyle, including diet, has caused many health problems, the so-called “diseases of civilization”, such as chronic metabolic diseases like obesity and diabetes, cardiovascular diseases, osteoporosis, cancer, etc. [1]. To address these issues, modern society has become more health-conscious, and many of today’s consumers are trying to adopt healthier eating habits and are looking for a proper and balanced diet to meet their nutritional needs [2]. In this context, the World Health Organization (WHO) strongly recommends reducing the intake of saturated fatty acids to 10% of the total energy intake for both adults and children [3]. When it comes to animal-based products, there is growing consumer interest in the welfare, safety and quality of these products [4,5,6]. Moreover, many consumers take into consideration the environmental impact of their food choices [6].

The poultry industry has experienced rapid global growth in recent years, driven by increasing consumer demand for poultry eggs and meat. Eggs are an inexpensive source of animal protein. They are widely regarded as a nutrient-rich, complete food, offering substantial health benefits [7]. To meet consumer demands and food market targets for nutritious and environmentally friendly value-added poultry products that promote animal welfare, researchers are focusing on supplementing poultry diets with innovative and advantageous feed additives to improve poultry production while safeguarding birds’ health and welfare [8,9]. Multiple studies have demonstrated that the fatty acid (FA) profile of eggs can be modified by adjusting the diets of laying hens with the inclusion of various byproducts rich in FAs and antioxidants [10,11,12,13,14].

Olive pulp (OP), a byproduct of the olive oil extraction process, has emerged as a valuable plant-based feed additive. In recent years, there has been a growing interest in using agricultural byproducts as animal feed, in line with the modern circular economy approach [15]. Using agro-industrial byproducts as alternative feed resources to feed livestock could also be a viable strategy to reduce feed costs, which account for up to 70% of total production costs, and save traditional feed sources such as grain for human consumption [16]. Research shows that OP is more than just a byproduct; it contains functional ingredients like essential FAs, polyphenols, oleuropein and tyrosol, which can be used to produce high-quality foods, extend their shelf life and improve their nutritional quality [15,17]. In addition, this waste recovery process meets consumer and societal expectations for safe, high-quality processed foods while reducing waste and minimizing the environmental impact [18].

In support of the above, many investigations show that incorporating OP into livestock diets ameliorates the FA profile by decreasing saturated fatty acids (SFAs) and increasing unsaturated fatty acids (UFAs) in the final product, such as meat from broilers [19,20,21,22], pigs [23], rabbits [24,25], lambs [26] and beef and dairy products [27,28,29,30,31]. Some of these reports also showed a net improvement in the health lipid indices of the final product, such as the atherogenic index (AI), the thrombogenic index (TI) and the ratio of hypocholesterolemic to hypercholesterolemic FAs (h/H) [24,29,30,31]. There is also evidence of reduced cholesterol levels in the final product [21], as well as increased total phenolics and in vitro antithrombotic properties [21,32].

A literature review provides very limited but encouraging data regarding the supplementation of OP in layers’ diets and its impact on the nutritional value of eggs. Existing research shows an improvement in the FA profile in favor of UFAs [16,33,34], an improvement in health lipid indices [34], and a reduction in the cholesterol concentration of eggs [16,33,35]. In their study, Iannaccone et al. [35] found that dried olive pomace, when supplemented in laying hens’ diet, affected gene expression, and this was directly correlated with reduced cholesterol in eggs. The majority of these feeding trials have been carried out in caged systems or experimental units and have focused on the impact of OP on the nutritional composition of eggs at a specific age. However, it is well known that, in addition to diet, a hen’s age affects the cholesterol, FA composition, and health lipid indices of its eggs [36,37,38]. Regarding housing system considerations, future studies should also take into account the recent EFSA report recommending cage-free housing for laying hens [39]. Taking advantage of existing knowledge on the beneficial effect of OP on the nutritional value of eggs and extending our previous work [34], we conducted the current study in cage-free-housed laying hens. The aim of this study was to investigate the dietary effect of dried olive OP on the FA profile, cholesterol and triglyceride content, total phenolics, oleuropein and hydroxytyrosol content, and health lipid indices of eggs produced by mid- and late-laying hens. Our findings could provide valuable insights into future feeding strategies in the poultry sector for the production of environmentally and welfare-friendly, functional, healthy eggs that meet the demands of today’s conscientious consumers.

## 2. Materials and Methods

### 2.1. Ethics Statement

The Research Ethics Committee of the Hellenic Agricultural Organization DIMITRA approved the experimental protocol of this study and the animal care procedures used (61366/11 November 2022).

### 2.2. Experimental Design, Animals and Diets

This research was carried out on a Greek commercial poultry farm. Dedousi et al. [40] provide a detailed description of the experimental design, the hens and OP used, and the ratios formulated. In brief, 180 Isa-Brown layers at 20 weeks of age were allocated to 15-floor littered pens covered with wood shavings (12 hens/pen) under the EU Directive 1999/74/EC on alternative housing systems for laying hens [41] and were fed with a basal layer diet. Environmental parameters, like the ventilation and lighting program, temperature and humidity, were automatically controlled according to breeding recommendations. Following an adaptation period of 3 weeks, when the hens were 23 weeks old, they were divided into three dietary groups, namely CON, OP4, and OP6, with 60 hens per group, 5 replicate-floor pens per group, and 12 birds per replicate-floor pen, according to the incorporation rate of OP in their diet. The control group (CON) was fed a maize and soya meal-based layer diet without OP, similar to the basal diet that was offered to the birds in the adaptation period. The OP groups received the CON diet supplemented with 4% and 6% olive pulp (OP4 and OP6). The dried OP used in this study was a commercial product in the form of flour, which was used to replace a certain amount of the soya meal and soya oil in the CON diet. Thus, three isonitrogenous and isocaloric diets were formulated. The ingredient composition and the nutritional analysis of the experimental diets (%), as well as the nutritional analysis and fatty acid content of the olive pulp used in the trial, were presented in our former work [40] and are shown in Appendix A. The nutritional analysis of the olive pulp and raw feed materials was conducted using the methods described by Suzanne Nielsen [42]. During the 36-week trial, production and egg quality traits, as well as biochemical indicators and the gut microbiota of the laying hens, were evaluated [40].

### 2.3. Determination of Egg Fatty Acid Profile, Cholesterol and Triglyceride Content, Total Phenolics, Oleuropein and Hydroxytyrosol Content, and Health Lipid Indices

A random sample of 150 eggs from laying hens aged 39 and 59 weeks was taken at each time period (50 eggs/group; 10 eggs/ floor pen; 300 eggs in total) and sent to an ISO-certified commercial laboratory for analysis. At each time point, eggs were collected on the same day for all groups. The content of polyphenols, cholesterol, triglycerides, oleuropein and hydroxytyrosol and the FA profile of the eggs were determined. At each time point, 10 final samples/group were formed after mixing and homogenizing 5 eggs from a total of 50 collected for each treatment group. The 10 samples of eggs formulated were double-analyzed to evaluate the above-mentioned parameters in the whole edible part of the egg (white and yolk). Saturated (SFAs), monounsaturated (MUFAs) and polyunsaturated fatty acids (PUFAs) were also assessed in the whole edible part of the egg (white and yolk).

#### 2.3.1. Determination of Egg Fat and Fatty Acid Profile

The FA content was determined via the Soxhlet extraction method (Soxtherm SOX412-MACRO by Gerhardt, Königswinter, Germany) [42], followed by a gravimetric calculation. A working mass of 10–15 g was placed in a round flask with 4N hydrochloric acid and brought to the boil for approximately 1 h, followed by filtration and drying at 45 °C for 10 h. The filter was placed in a pre-weighed glass extraction beaker with petroleum ether for Soxhlet extraction, and then the beaker was weighed. The fat percentage was calculated from the difference between the glass extraction beaker containing the sample and the empty glass from the mass of the sample that was weighed multiplied by 100.

The extracted FAs were trans-esterified in a methanolic potassium hydroxide solution, and the FAMES samples were analyzed by gas chromatography (GC-FID) [42]. All reagents used were of GC grade and were purchased from Sigma-Aldrich (Merck KGaA, Darmstadt, Germany). Chromatographic analyses were carried out with a Shimadzu GC-2010 Plus High-End gas chromatography system (Shimadzu Europa GmbH, Duisburg, Germany), equipped with a Flame Ionization Detector (Shimadzu Europa GmbH, Duisburg, Germany) (FID). The column used was a Supelco SP2560 (Merck KGaA, Darmstadt, Germany), measuring 100 m × 0.25 mm × 0.20 μm. Helium (grade 99.999%) was used as a carrier gas at a flow rate of 2 mL/min. The injection volume was 1 μL, with a split ratio of 1:50, and the injector temperature was 250 °C. The detector temperature was set at 250 °C. The temperature program employed was as follows: the initial oven temperature was 110 °C (7 min), and then it was increased at 3 °C/min to 190 °C (2 min); then, in the first step, it increased at 0.5 °C/min to 205 °C; in the second step, it increased at 5 °C/min to 230 °C (5 min); and in the third step, it increased at 5 °C/min to a final temperature of 240 °C for 5 min. The total run time was 82.67 min [34].

#### 2.3.2. Total Phenolics

The phenolic acid content of the eggs was determined by an ultraviolet–visible (UV-VIS) spectrophotometry method using Folin–Ciocalteu reagent 2N (Merck KGaA, Darmstadt, Germany). Approximately 5 g of each sample was used, and a certain amount of methanol was used for the following sample extraction and filtering. Afterwards, 5 mL of 10% Folin–Ciocalteu reagent was added to the filtrate solution with 4 mL of 7.5% sodium carbonate, and the tubes were stern. After 60 min, the optical density was measured with 10 mm glass cells at 765 nm by the UV-VIS method. A standard solution of 1000 ppm gallic acid was used for the calibration curve, and the total phenolic acid content was expressed as mg/kg (ppm) [34].

#### 2.3.3. Determination of Cholesterol

The determination of cholesterol was based on a method in accordance with Commission Regulation (EEC) No 2568/91 [43]. Approximately 5 g of each sample was used for cholesterol determination. The different alcoholic compound fractions were separated from the unsaponifiable matter by thin-layer chromatography (TLC) on a basic silica gel plate. The fractions collected from the unsaponifiable matter using TLC were derivatized into trimethylsilyl ethers and analyzed by capillary column gas chromatography with a split injection and flame ionization detector. Chromatographic analyses were carried out with a GC equipped with an FID. The injection volume ranged from 0.5 to 1 μL, with a split ratio of 1:50 to 1:100, and the injector temperature was 280–300 °C. The detector temperature was set at 280–300 °C. The linear velocity of the carrier gas was as follows: helium: 20 to 30 cm/s and hydrogen: 30 to 50 cm/s. The oven temperature was set at 260 ± 5 °C Isothermal. The percentage of sterols was calculated from the ratio of the relevant peak area to the total peak area for sterols [34].

#### 2.3.4. Determination of Triglycerides

The triglyceride content was determined according to International Standard ISO 18395:2005 [44]. The sample (about 100 mg of sample) was dissolved in tetrahydrofuran (THF) and analyzed by gel permeation chromatography (GPC) with THF as the mobile phase. Detection was carried out using a refractive index detector, and the content was expressed as a mass fraction (grams per 100 g).

#### 2.3.5. Determination of Oleuropein and Hydroxytyrosol

The determination of oleuropein and hydroxytyrosol was conducted through chromatographic analysis, following the guidelines of the International Olive Council [45]. Specifically, 2–3 g of sample was used for oleuropein and hydroxytyrosol determination. Chromatographic analyses were carried out with high-performance ternary-gradient liquid chromatography (HPLC), equipped with a C18 reverse-phase column (4.6 mm × 25 cm) of type Spherisorb ODS-2 5 µm, at 100 A°, with a spectrophotometric UV detector at 280 nm and an integrator at room temperature. Spectral recording for identification purposes was facilitated by using a photodiode detector with a spectral range from 200 nm to 400 nm. Reagents should be pure HPLC grade. Oleuropein and hydroxytyrosol contents were expressed in mg/kg and calculated by measuring the sum of the areas of the related chromatographic peaks.

#### 2.3.6. Determination of Health Lipid Indices

On the basis of the proportions of specific FAs and their groups (saturated fatty acids: SFAs; monounsaturated fatty acids: MUFAs; polyunsaturated fatty acids: PUFAs), certain health indicators of egg FAs were determined. These include the atherogenic index (AI), the thrombogenic index (TI) and the h/H index, which expresses the ratio of hypocholesterolemic to hypercholesterolemic FAs. These indicators were calculated using the following equations:

Atherogenic index [46]:AI = (4 × C14:0 + C16:0 + C18:0)/(ΣMUFA + ΣPUFA-n-6 + ΣPUFA-n-3)
where Σ = Summatory.

Thrombogenic index [47]:TI = (C14:0 + C16:0 + C18:0)/(0.5 × ΣMUFA + 0.5 × ΣPUFA-n-6 + 3 × ΣPUFA-n-3 +ΣPUFA-n-3/ΣPUFA-n-6)

Hypocholesterolemic-to-hypercholesterolemic FAs ratio [48]:h/H = C18:1n9c + C18:2n6c + C18:3n3c + C18:3n6c + C20:2n6 + C20:3n6 + C20:4n6 + C22:6n3/C14:0 + C16:0 

### 2.4. Statistical Analysis

The statistical analysis of data was performed with the use of Jeffreys’ Amazing Statistics Program JASP software (v 0.16.4; JASP Team, 2022) [49]. The Shapiro–Wilk test was employed to test the normality of the data, and the Levene’s test was used for the homogeneity of variance assessment. The group and age of the birds were used as fixed factors to compare the average values of the parameters evaluated between the treatment groups at 39 and 59 weeks of age. Post hoc evaluation was conducted with Tukey’s test. Non-parametric Kruskal–Wallis and Mann–Whitney tests were used for comparisons when the distribution was not normal. The significance level was set at *p* ≤ 0.05 for all comparisons.

## 3. Results

### 3.1. Egg Cholesterol and Triglyceride Content, Total Phenolics, Oleuropein and Hydroxytyrosol Content, Concentration of Saturated (SFAs), Monounsaturated (MUFAs) and Polyunsaturated (PUFAs) Fatty Acids, and Health Lipid Indices

The findings of the present study showed that the polyphenol, cholesterol, triglyceride and oleuropein contents of eggs were significantly (*p* < 0.05) affected by the age of the hens (Table 1). Specifically, with increasing hen age, there was a significant increase (*p* < 0.05) in the eggs’ polyphenol and cholesterol contents, while the triglyceride and oleuropein concentrations decreased significantly (*p* < 0.05). Furthermore, it was observed that the inclusion of OP in the birds’ feed significantly affected the cholesterol concentration in the eggs produced, as the cholesterol levels in the eggs from the OP groups were significantly lower (*p* < 0.05) than those observed in the CON eggs. The same significant (*p* < 0.05) group effect was observed in eggs from 59-week-old birds but not in those from younger hens, where the differences among groups were numerical (*p* > 0.05). In addition, the analysis of eggs from the first sampling period (week 39) revealed that the OP eggs had a higher (*p* < 0.05) concentration of polyphenols compared to the CON eggs. However, the observed difference was only significant (*p* < 0.05) between the CON and OP6 groups. On the other hand, in the second sampling period (week 59), the polyphenol content of the eggs did not differ between the groups (*p* > 0.05). The concentration of hydroxytyrosol in the eggs from all the groups was below the detection limit (DL) of 3 ppm.

The data analysis showed that bird age significantly (*p* < 0.05) affected the eggs’ SFA and MUFA concentrations (g/100g fat). As Table 1 shows, the SFA content of the eggs decreased significantly (*p* < 0.05) with the age of the birds. This age effect was statistically significant (*p* < 0.05) for the groups that consumed olive paste in their diet (OP) and numerical in the CON group (*p* > 0.05). On the other hand, the MUFA content of the eggs increased significantly with bird age (*p* < 0.05). However, this significant effect (*p* < 0.05) of age on MUFA concentration was only observed in the OP eggs. Furthermore, the eggs’ MUFA concentration was significantly affected by the group (*p* < 0.05) and by the interaction of group × age (*p* < 0.05). More specifically, the OP4 eggs had significantly higher MUFA concentrations than the other two groups (*p* < 0.05). As observed in the first sampling period (week 39), the OP6 eggs had the lowest MUFA concentration among the groups (*p* < 0.05). However, in the second sampling period, the CON eggs had a significantly lower (*p* < 0.05) MUFA concentration compared to the OP eggs.

The PUFA content of the eggs was significantly affected by both the group (*p*< 0.05) and the group × age interaction (*p* < 0.05). In particular, the highest concentration of PUFAs was observed in the OP6 eggs, and the lowest concentration was observed in the OP4 eggs. These observed differences between the OP6 and OP4 groups were statistically significant (*p* < 0.05). The CON eggs had a similar PUFA concentration to the OP6 eggs (*p* > 0.05) and a statistically higher PUFA content than the OP4 eggs (*p* < 0.05). In the eggs collected in the first sampling period (week 39), the concentration of PUFAs was statistically significantly higher in the OP6 group compared to the other groups (*p* < 0.05). In the second sampling period (week 59), the PUFA content of the eggs from the OP4 group was significantly lower than in the other groups (*p* < 0.05), whereas this parameter was not significantly different between the eggs from the CON and OP6 groups (*p* > 0.05). Furthermore, as shown in Table 1, the eggs’ PUFA content increased significantly with bird age in the CON group (*p* < 0.05). Moreover, in the OP6 group, the PUFA concentration decreased with increasing hen age (*p* < 0.05), whereas in the OP4 eggs, the PUFA concentration did not change between sampling periods.

The PUFA/SFA ratio was significantly affected by the group (*p* < 0.05) and by the group × age interaction (*p* < 0.05; Table 1). This ratio was significantly higher (*p* < 0.05) in the CON eggs and the OP6 eggs than in the OP4 eggs. In addition, in the eggs analyzed in the first sampling period (week 39), the PUFA/SFA ratio was significantly higher in the OP6 group than in the OP4 group (*p* < 0.05) and numerically higher than in the CON group (*p* > 0.05). Finally, a significant increase in the PUFA/SFA ratio was observed between the samples in the eggs from the CON group (*p* < 0.05). These significant differences in the eggs’ concentrations of PUFAs and SFAs between groups at different ages and within each group between ages resulted in relative differences in the PUFA/SFA ratio.

The percentage of n6 and n3 PUFAs in the eggs was significantly influenced by the group (*p* < 0.05; n6 PUFAs), age (*p* < 0.05; n3 PUFAs) and the interaction of group × age (*p* < 0.05; n6 PUFAs, n3 PUFAs; Table 1). The eggs from the OP4 group had a significantly lower percentage of n6 PUFAs than the eggs from the other groups (*p* < 0.05). In the eggs from the younger birds (week 39), the percentage of n6 PUFAs in the OP6 eggs was significantly higher (*p* < 0.05) than that found in the other groups. In the second sampling period, the lowest percentage of n6 PUFAs was recorded in the OP4 eggs, but significant differences were only observed between the CON and OP4 groups (*p* < 0.05). Furthermore, our results showed that the percentage of n6 PUFAs in the OP6 eggs decreased significantly with the increasing age of the birds (*p* < 0.05). As shown in Table 1, the n3 PUFA content of the eggs from all the groups increased significantly with the increasing age of the birds (*p* < 0.05). In addition, when eggs from the 59-week-old hens were analyzed, it was found that the n3 PUFA content of the CON eggs was significantly higher than that of the OP groups (*p* < 0.05). The n6 PUFA/n3 PUFA ratio in the eggs of all the groups decreased significantly between the sampling periods (*p* < 0.05). This was mainly due to the significant increase in the n3 PUFA percentage with the increasing age of the birds (*p* < 0.05). In addition, the n6 PUFA/ n3 PUFA ratio in the OP4 eggs was significantly lower than in the OP6 eggs (*p* < 0.05) and numerically lower than in the CON eggs (*p* > 0.05). The same significant group effect was observed in the first sampling period of the eggs (*p* < 0.05), but not in the second sampling period (*p* > 0.05).

The data analysis (Table 1) showed that bird age had a significant effect on egg health lipid indices (*p* < 0.05). Specifically, as the birds aged, the AI and TI decreased significantly, and the h/H ratio increased (*p* < 0.05). For the TI and h/H indices, this significant effect of age was observed in all the experimental groups. However, the observed decrease in the AI with increasing bird age was significant for the OP groups (*p* < 0.05) and numerical for the CON group (*p* > 0.05). The results of the present study showed that the egg health lipid indices were not affected by the group (CON, OP4, OP6) or the group × age interaction.

### 3.2. Total Fat Content % and Fatty Acid Composition

Our findings revealed a significant age effect on the total fat content % of the eggs (*p* < 0.05). The oldest hens produced eggs with a higher fat content % than the youngest birds (Table 2). Significant differences in the fatty acid composition of the eggs (g/100 g of fat) were found both between the groups and between the sampling periods, which are shown in detail in Table 2. Of the individual FAs isolated from the eggs during the two sampling periods, only myristic acid (C14:0) and cis-8,11,14-eicosacetrienoic acid (C20:3n6) were not affected by group (OP, CON; *p* > 0.05), age (*p* > 0.05) or the interaction of group × age (*p* > 0.05).

In general, palmitic acid (C16:0) was the most abundant SFA detected in the eggs from all the dietary groups in both sampling periods (about 25.79%), followed by stearic acid (C18:0) at about 5.72%. According to Table 2, the concentration of palmitic acid (C16:0) in the eggs from all the groups decreased significantly with the increasing age of the birds (*p* < 0.05). The concentration of stearic acid (C18:0) remained constant between the sampling periods in the CON and OP4 eggs (*p* > 0.05) and decreased significantly in the OP6 eggs (*p* < 0.05). In addition, in the first sampling period (week 39), the eggs from the OP6 group had a significantly (*p* < 0.05) higher level of stearic acid (C18:0) compared to the other groups. Based on our results, oleic acid (C18:1n9c) was the most abundant of the MUFAs found in the eggs from all the groups studied in both sampling periods, accounting for approximately 51.5% (Table 2). The concentration of oleic acid in the eggs increased significantly (*p* < 0.05) with the increasing age of the birds. However, it was only in the groups consuming OP that this significant effect of age (*p* < 0.05) on oleic acid (C18:1n9c) concentration in the eggs was observed. As indicated by the group means, the highest oleic acid concentration was found in the eggs from the OP4 group, followed by the eggs from the CON and OP 6 groups. However, significant differences were only observed between the OP4 and OP6 groups (*p* < 0.05). Moreover, in the first sampling period (week 39), the OP6 eggs had significantly lower oleic acid levels than the other groups (*p* < 0.05). In both sampling periods, the second most abundant MUFA in the eggs from all the groups analyzed was palmitoleic acid (C16:1), accounting for approximately 4.44%. There was a significant decrease (*p* < 0.05) in the concentration of this acid between the sampling periods (Table 2). However, the same significant effect of age (*p* < 0.05) was only observed in the CON group. According to the group mean, the OP4 eggs had a significantly higher palmitoleic acid concentration than that recorded in the eggs from the other groups (*p* < 0.05), followed by the OP6 and CON eggs. In the first sampling period, the lowest concentration of palmitoleic acid was found in the eggs from the OP6 group and the highest was found in the eggs from the OP4 group (*p* < 0.05). However, in the second sampling period, the eggs from the OP groups had a significantly higher concentration of palmitoleic acid compared to the CON group (*p* < 0.05).

As shown in Table 2, among the PUFAs, linoleic acid (C18:2n6c) was the most abundant in the eggs from all the groups studied in both sampling periods, accounting for about 10.75%. The data analysis showed that the eggs from the OP4 group had significantly lower levels of linoleic acid than the other groups (*p* < 0.05). When the results were examined by sampling, it was found that at 39 weeks of age, the eggs from the OP6 group had a significantly higher linoleic acid concentration than the other groups (*p* < 0.05). However, in the second sampling period, the linoleic acid content of the eggs did not differ between the CON and OP6 groups (*p* > 0.05), whereas the eggs from the OP4 group had a significantly lower linoleic acid concentration compared to the eggs from the CON group (*p* < 0.05). The second most abundant fatty acid of the PUFAs detected in the eggs from all the groups in both sampling periods was Cis-11,14,17-eicosatrienoate acid (C20:3n3) at about 0.323%. The concentration of this fatty acid increased with the age of the hens (*p* < 0.05). This significant age effect (*p* < 0.05) was only noted in the CON group though. Finally, in the second sampling period, significant differences in the eggs’ concentration of C20:3n3 were observed among the groups. In particular, the CON eggs had significantly higher levels of C20:3n3 than the OP6 eggs (*p* < 0.05), whereas numerical differences were observed between the OP eggs (*p* > 0.05) and between the CON and OP4 eggs (*p* > 0.05).

## 4. Discussion

This study showed that the addition of 4% and 6% OP in the diet of laying hens increased the total phenolic concentration in their eggs, especially at the highest dose of 6%, confirming our previous findings [34]. This result could be attributed to the higher concentration of total polyphenols in the experimental diets compared to the CON group (123.06 ± 37.66 ppm, 137.24 ± 24.76 ppm, 95.40 ± 23.80 ppm for OP4, OP6, and CON groups, respectively; Appendix A). Previous studies have shown that feeding polyphenol-rich diets to laying hens improves the concentration of total phenolics in their eggs [14,50,51]. Currently, there is a lack of research evidence on the effect of OP on the concentration of total phenolics in eggs. Similarly, studies carried out in broilers showed that the supplementation of their feed with 5% OP increased their breast meat’s total phenolic content [32]. Accordingly, Ibrahim et al. [21] found that the inclusion of 15% and 30% fermented or enzymatically fermented olive pomace in broiler diets significantly increased the total phenolic and flavonoid contents of breast meat compared to controls. Our analysis revealed that the beneficial dietary effect of OP on the total phenolic content of eggs was only observed in eggs produced by younger hens. Later in the production cycle, an increase in the total polyphenol content of the eggs was observed in all the groups, eliminating the previously observed differences. To the best of our knowledge, there are no data in the current literature on the effect of hen age on egg phenolic content to compare our results to. Current research indicates that diets high in polyphenols provide long-term protection against neurodegenerative diseases, certain types of cancer, cardiovascular disease, gastrointestinal disorders, osteoporosis, type 2 diabetes, lung damage, and pancreatitis [52]. Consequently, the increased levels of egg phenolics observed with OP supplementation in the diets of younger hens or with the natural aging of birds in all diet groups are of great interest to consumers due to their potential health benefits.

Hydroxytorosol and oleuropein are known to be part of the phenolic profile of OP [53]. In the present study, we therefore investigated the dietary effect of OP on the concentration of these bio-compounds in eggs. Our results showed that the concentration of hydroxytyrosol in the eggs from all the groups was below the detection limit (DL) of 3 ppm. This finding was quite reasonable for the OP groups, as the hydroxytyrosol content of the OP used in the present study was also less than 3 ppm (Appendix A). The oleuropein egg content was similar among groups, indicating no effect of OP on this parameter when added to laying hens’ diet at the studied levels of 4% and 6%. Even though phenolic compounds found in poultry feed can be transferred and deposited in egg yolk as mentioned above, the conditions and factors affecting their bioavailability and deposition in egg yolk are still being investigated. It has previously been documented that it is the structure of the polyphenols, rather than their concentration, that determines the rate and extent of absorption, as well as the nature of the circulating metabolites in plasma [54]. In many cases, the parent polyphenol cannot be detected because phenolic antioxidants are usually metabolized into completely different molecules. As a result, it is difficult to track the fate of each compound in the body, as many new compounds may be formed during the metabolism of each phenolic compound [55]. Moreover, recent reports have shown that the levels of polyphenols in different tissues of the body are not directly related to their dietary intake [56]. This is also supported by the findings of Papadopoulos et al. [57], who added an olive leaf extract that was rich in oleuropein to the diet of laying hens at incorporation rates of 1% and 2.5% and observed that the highest total egg phenolic content was obtained with the lowest dose (1%) of oleuropein leaf extract. On the other hand, our study showed that the concentration of oleuropein in the eggs was only influenced by the age of the hens. In particular, the oleuropein concentration decreased with hen age in the eggs from all the groups. However, it is not possible to compare our results with other reports due to the lack of research on the dietary effect of OP or the age of the hen on eggs’ oleuropein content.

Our research has shown that including OP in the diet of laying hens leads to a reduction in cholesterol levels in their eggs, which was found to be significant in the second phase of the laying period. A medium-sized egg (50 g) is estimated to contain approximately 186 mg of cholesterol [58]. In our study, the cholesterol content per egg, based on the average egg weight, as described in our previous work [40], was approximately 190.73 mg for the CON eggs and 166.80 mg and 172.53 mg for the OP4 and OP6 eggs, respectively. Given that a dietary intake of high levels of cholesterol is associated with an increased risk of cardiovascular disease [58], the results of this study are particularly important as they may lead to significant health benefits for consumers. Such a beneficial dietary effect of OP on egg cholesterol might be related to the phenolic compounds and/or high fiber content of OP, as documented by other authors. In particular, Iannaccone et al. [35] found that the addition of 10% dried OP in the diet of 20-week-old laying hens for 28 days significantly decreased yolk cholesterol. These researchers suggested that the obtained result was due to OP polyphenolic compounds modulating several genes belonging to cholesterol biosynthetic pathways. In a similar study, Hashish and Abd El-Samee [33] associated the lowering effect of olive cake (OC) plus barley malt rootlets on egg yolk cholesterol concentrations with the high crude fiber and oil content of OC. In line with our findings, Safwat et al. [16] observed that feeding 31-week-old laying hens with 7% OC for 12 weeks reduced their eggs’ cholesterol. Crude fiber content, PUFA content, and OC phenolic compounds such as flavonoids and tannins were associated with the hypocholesterolemic effect of OC. Numerous studies have shown that feeding polyphenol-rich diets [59,60,61,62,63,64] or high-fiber diets [62,65,66,67,68] to laying hens improves egg cholesterol, but the mechanisms involved in this mode of action are still under investigation. Research suggests that flavonoids, which are naturally occurring compounds with multiple phenolic structures, inhibit adipogenesis, promote lipolysis and apoptosis in adipose tissue cells, and reduce cholesterol deposition in egg yolk and therefore have the potential to affect fat deposition in poultry at different ages and stages of production [60], as well as to improve eggs’ nutritive value [69].

Dietary fiber is also thought to play a role in cholesterol metabolism and egg quality in poultry by reducing cholesterol absorption, binding bile salts in the intestine, shortening the intestinal transit time and increasing fecal sterol excretion [16,68,70,71]; however, the underlying mechanisms remain unclear. Within this context, many studies suggest that the reduction in cholesterol levels and the positive effects on egg quality may be associated with microbial metabolites that can modulate lipid metabolism to influence egg quality [72], in particular, with short-chain fatty acids (SCFAs) produced by the microbial degradation of dietary fibers [73,74]. In a recent study [40], we found a link between dietary OP exposure and a shift in the microbial composition of chicken feces towards the genera *Megasphaera* and *Megamonas*, which are involved in the fermentation of dietary fiber and carbohydrates and produce SCFAs. Based on the above and the results of the present study, we hypothesized that the inclusion of OP in the diet of laying hens could lead to a reduction in cholesterol levels in their eggs by regulating the composition of the microbiome and subsequently the production of microbial metabolites such as SCFAs.

The current study showed a significant age effect on egg cholesterol levels, confirming previous reports. Consistent with our results, Abdalla and Ochi [36] found that as laying hens aged from 22 to 38 weeks, total egg cholesterol increased from 178.04 to 214.75 mg/egg. In contrast, Johansson [75] reported the opposite (decreasing) age effect on egg cholesterol levels as hens’ age progressed from 32 to 72 weeks. According to Ko et al. [48], eggs from young (24 weeks), intermediate (42 weeks) and older (74 weeks) hens had similar cholesterol levels. A practically stable concentration of egg cholesterol throughout the laying period has also been reported by other authors, although they noted a numerical increase with hen age [37,76,77]. The inconsistency in the results of these studies with regard to the effect of hen age on egg cholesterol content could be due to differences in a number of factors that have been shown to influence this parameter, such as the hens’ breed, line, diet, housing system, etc. [37,75,78].

Consistent with former studies, we found that the concentration of egg triglycerides decreased as the hens aged [79]. On the other hand, our results demonstrated that supplementing the hens’ diets with OP did not affect the eggs’ triglyceride concentrations. Contrary to our observations, a reduction in egg triglyceride content has been documented in the limited literature available [16,33]. These authors attributed the improvement in this parameter to the high fiber, PUFA and polyphenol content of OP. This view is supported by existing research on feeding fiber- and polyphenol-rich diets to laying hens [62,63,68].

The analysis of the FA profile of eggs in this study showed that UFAs accounted for 67.94% and SFAs accounted for 32.07% in the eggs from all the groups, favoring the nutritional value of the eggs. This is in line with the literature, showing that around 60–70% of the FAs in eggs are unsaturated [38,80,81]. When individual FAs were considered, palmitic, oleic and linoleic acids predominated in the egg content regardless of OP supplementation or hen age, confirming previous reports [34,82]. According to our results, the SFA concentration in the eggs was only affected by the age of the hens and not by the addition of OP in the birds’ diet. In particular, we found that the percentage of SFAs decreased with increasing hen age for all the dietary treatments, but this was more pronounced in the OP eggs. This finding can be mainly attributed to the decreasing concentration of palmitic acid, the most abundant SFA recorded in all the groups, with increasing hen age. A similar reduction in the concentration of SFAs in the eggs during the laying period has also been reported by other researchers [38,48,81].

In our previous investigation [34], carried out with caged laying hens, we found a decrease in the percentage of SFAs as the OP incorporation rate in the hens’ diet increased. We mainly attributed this finding to a lowering effect of OP on palmitic acid concentration and, to a lesser extent, on stearic and myristic acid concentrations. However, in the present study, the levels of these FAs were not affected by OP supplementation, but the observed reduction was due to the increasing age of the laying hens during the laying period. In a study by Safwat et al. [16], the palmitic and stearic acid concentrations remained unaffected, whereas myristic acid decreased, despite the observed reduction in SFA levels in the eggs produced by hens fed 7% olive cake compared to the controls. A similar reduction in egg SFA concentration was reported by Hashish and Abd El-Samee [33] in their 12-week trial, in which 54-week-old hens were fed olive cake (OC) plus barley malt rootlets.

In this study, MUFA levels were influenced by both hen age and OP supplementation. With increasing hen age, the percentage of MUFAs in the CON eggs remained stable but increased in the OP eggs, resulting in a higher concentration of MUFAs in the OP groups compared to the CON group in the second phase of the production cycle. This result could be attributed to the proportional elevation in the oleic acid concentration in the OP eggs with hen age, which was not observed in the CON group. A similar increase in oleic acid concentration and, consequently, MUFA levels with hen age from 28 to 80 weeks old was also reported by Zita et al. [38]. On the other hand, other researchers have found a decrease in the MUFA content of eggs as the laying period progresses [48,81]. In our previous study, adding OP to hens’ diets at various levels, ranging from 2 to 6%, did not affect the MUFA levels in the OP eggs compared to the controls [34]. However, an increase in oleic acid concentration and MUFA egg content due to OP incorporation in laying hens’ diet has previously been documented [16,33]. In the current study, the lower levels of MUFAs in the OP6 eggs compared to the OP4 eggs in the first sampling period were due to similar observed differences in oleic acid levels between these groups. Consistent with the current findings, our previous study also found that the percentage of MUFAs in eggs produced by hens fed a higher dose of OP (5%) was lower than that found in eggs produced by laying hens fed a lower dose of OP (4%) [34].

Taking into account the PUFA content of the eggs, our results showed significant dietary effects of OP in each phase of the production cycle, as well as significant age effects within groups. The higher percentage of PUFAs recorded in the OP6 eggs compared to the other groups in the first phase of the laying period could be attributed to the higher concentration of linoleic acid observed in this group at this time. Consistent with our findings, an increasing effect of OP on linoleic acid levels and the percentage of PUFAs in eggs has been reported previously [16,33,34]. As in the current study, in our former, similar trial, we noticed that the increase in egg PUFA content observed with OP supplementation was dose-dependent, with better results obtained with higher incorporation rates (5% and 6%) compared to lower ones (2–4%) [34].

With increasing hen age, the PUFA levels increased in the CON eggs, remained constant in the OP4 eggs and decreased in the OP6 eggs. Similar changes in linoleic acid content (numerical for CON group and significant for OP6 group) within groups with hen age may explain the observed differences between the dietary treatments. The increase in PUFA content observed in the CON eggs with hen age is consistent with the findings of Lesic et al. [81] and partially consistent with the findings of Ko et al. [48]. More specifically, the latter authors found that the percentages of PUFAs in eggs obtained from young (24 wk) and intermediate (42 wk) hens were similar, but this percentage was found to be increased in eggs from late-laying (74 wk) hens. The decrease in the PUFA levels of OP6 eggs observed in older hens is in line with the findings of Zita et al. [38]. Looking at the FA profile of the OP eggs as a whole, it can be seen that in younger hens, the OP6 eggs had less MUFAs in favor of PUFAs compared to the OP4 eggs, taking into account the similar level of SFAs between the groups. However, as the hens aged, the PUFA content in the OP6 eggs decreased in favor of MUFAs, whereas the PUFA content in the OP4 eggs remained constant. As a result, in the second phase of the laying period, the CON and OP6 eggs had similar PUFA levels, which were higher than those found in the OP4 eggs. The increased concentration of PUFAs in the OP6 eggs detected in the first phase of the production cycle and the elevated levels of MUFAs in the OP eggs observed in the second phase of the laying period are beneficial from a nutritional point of view. The UFAs found in egg yolk have been shown to increase antioxidant capacity, have anticancer activity, format lipid membranes and regulate membrane-dependent progression [83]. Moreover, a diet rich in MUFAs has been associated with a reduction in low-density lipoprotein (LDL) cholesterol and an increase in high-density lipoprotein (HDL) cholesterol, as well as a blood pressure-lowering effect, which may be beneficial for health [84].

The PUFA/SFA ratio is commonly used to estimate the nutritional value of dietary fat. Egg yolk fats with a higher PUFA/SFA ratio may confer greater health benefits because dietary PUFAs have been shown to decrease low-density lipoprotein cholesterol and cholesterol levels in serum, while SFAs contribute to high serum cholesterol concentrations [85]. Thus, it has been shown that dietary PUFA/SFA ratios greater than 0.45 are considered safe for human consumption [86] and adequate to protect against developing ischemic heart disease [87]. In the current study, the PUFA/SFA ratio in the eggs from all the groups was slightly below 0.45, ranging from 0.33 to 0.40. However, it was higher than the value of 0.32 reported by Khan et al. [88] for table eggs and close to the value of 0.42 reported by Bondoc et al. [85] for hen eggs. From a nutritive point of view, in the OP groups, the optimal PUFA/SFA ratio was achieved with a higher incorporation rate of OP, confirming our former findings [34]. Taking into account the similar percentages of SFAs in the eggs from all the dietary treatments, the differences in the PUFA/SFA ratio observed between the groups in the first phase of the production cycle were due to the observed differences in the PUFA levels between the treatments. The improvement in the PUFA/SFA ratio in the CON eggs with hen age is attributed to the increase in PUFAs. Such a positive effect of age on this index has been reported previously [81]. On the other hand, the non-significant age effects on the PUFA/SFA ratio observed in the OP groups are consistent with the findings of Ko et al. [48] and Zita et al. [38].

The differences in n6 PUFA levels observed among the treatments in each phase of the production cycle and during the laying period are attributed to relevant changes in linoleic acid concentration, which was the most abundant n6 PUFA fatty acid and has already been analyzed in detail. According to our results, the percentage of n3 PUFAs increased with the age of the hens. This result was mainly due to the increase in a-linolenic acid concentration observed in all the dietary treatments and, to a lesser extent, to an increase in the eicosatrienoate acid (C20:3n3) concentration found only in the CON eggs. These changes in n3 PUFA concentrations with hen age resulted in the higher percentage of n3 PUFAs in the CON eggs compared to the OP eggs in the second phase of the production cycle. Consistent with our findings, Lesic et al. [81] observed that n3 PUFA concentrations in eggs increased from 0.26% to 0.50% as the hens’ age increased from 21 to 55 weeks. On the other hand, Ko et al. [48] reported a decrease in n3 PUFA concentration in eggs from early (24 weeks)- to intermediate (42 weeks)-laying hens, but later, the aforementioned value remained constant up to 74 weeks, ranging from 2.22% to 2.03%. A decrease in n3 PUFA concentrations from 1.64% to 1.30% was also reported by Zita et al. [38] in eggs analyzed at the beginning of the first (from the 28th to 30th week of age) and second (from the 78th to 80th week of age) laying cycles. Contrary to the present study, in our previous work carried out in caged laying hens, we found that at incorporation rates of OP ranging from 2% to 4%, as well as in the CON group, a-linolenic acid levels and, consequently, n3 PUFA concentrations were not detected. However, the addition of higher levels of OP (5% and 6%) to the hens’ diets resulted in a concomitant increase in a-linolenic acid levels and, consequently, n3 PUFA concentration, with better results obtained at the 5% dose. Similarly, in a 12-week trial by Safwat et al. [16], the supplementation of laying hens’ diets with 7% OP increased a-linolenic acid levels compared to CON eggs.

Another common index used to evaluate the nutritional value of fats is the n6 PUFA/ n3 PUFA ratio, with an ideal ratio of between 3:1 and 5:1 [83]. n3 PUFAs are precursors of eicosanoids, which play an important role in regulating inflammation. The eicosanoids derived from n6 FAs are pro-inflammatory, while those derived from n3 FAs are anti-inflammatory [83]. This study showed an improvement in the ratio n6/n3 FAs with hen age, indicating that eggs produced by older hens had better nutritional value. Such a positive (decreasing) effect of hen age on this index was also reported by Yannakopoulos et al. [89]. In contrast to our observations, Ko et al. [48] found that eggs from middle- and late-age-laying hens had similar n6 PUFA/n3 PUFA ratios (22.2 and 22.3, respectively). A non-significant age effect on this index has also been reported by other researchers [38,81]. According to our results, in the first phase of the production cycle, the most favored n6 PUFA/n3 PUFA ratio between OP groups was recorded in the OP4 eggs, mainly due to the lower levels of n6 FAs found in this group. As stated by Hashish and Abd El-Samee [33], feeding 28.5 g of olive cake (OC) plus 51.1 g of barley malt rootlets (BMR) or 57 g of OC plus 25.5 g or 51.1 g of BMR increased the n6 PUFA/n3 PUFA ratio from 8.6 to 18 compared to the controls (2.6). In contrast, the addition of 7% OP to the hens’ diet improved this index from 18.92 in the CON group to 15.03 in the OP group [16]. The ratio of n6 PUFAs to n3 FAs in eggs is 12:1 or even higher [81,90], exceeding the recommended levels, which is consistent with our findings.

While the PUFA/SFA ratio is widely used to assess the nutritional quality of dietary fats, it is often considered too broad and inadequate to assess the atherogenic potential of a food [91], as individual SFAs and PUFAs may have different metabolic effects [87]. FAs can either contribute to or help prevent atherosclerosis and coronary thrombosis, depending on their impact on serum cholesterol and low-density lipoprotein (LDL) cholesterol levels [61]. Therefore, the atherogenic index (AI) has been developed to better assess the atherogenic effect of a food [47]. Among SFAs, C14:0 and C16:0 are highly atherogenic, while C18:0 is generally neutral in terms of atherogenicity but is considered thrombogenic [61,91]. In contrast, UFAs are thought to be anti-atherogenic, helping to prevent plaque build-up and reduce phospholipids, cholesterol and esterified FAs [46,92]. Consequently, consuming foods with a lower AI may help reduce total cholesterol and LDL cholesterol in the blood [93]. The thrombogenic index (TI) is another metric often used in FA composition studies to assess thrombogenicity [91]. It represents the clot-forming potential of FAs and indicates the likelihood of blood clots forming in vessels. It also shows the contribution of various FAs, highlighting the balance between pro-thrombogenic FAs (C12:0, C14:0 and C16:0) and anti-thrombogenic FAs, including MUFAs and the n3 and n6 FA families [47]. Studies suggest that animal products with a low TI may reduce the risk of atrial fibrillation [94]. In summary, both the AI and TI can help evaluate the impact of FA composition on cardiovascular health (CVH). Regarding the h/H index, it expresses the ratio of hypocholesterolemic to hypercholesterolemic FAs. Santos-Silva et al. [95] suggest that a higher ratio of hypocholesterolemic to hypercholesterolemic FAs makes an oil or fat more suitable for human nutrition. Thus, a lower AI and TI and higher ratios of h/H FAs are indicators of better nutritional quality for fats and may reduce the risk of coronary heart disease (CHD), although no organization has established recommended values for these indices yet [47,91].

The current research showed that the supplementation of OP in the diet of laying hens at both levels studied (4% and 6%) did not affect health lipid indices. On the other hand, the AI, TI and h/H ratio improved significantly with the hen’s age in all the dietary treatments, suggesting that the eggs produced by older hens had superior nutritional value. To the best of our knowledge, the existing research data on the effect of dietary OP or hen age on the health lipid indices of the produced eggs are limited. Consistent with our study, Zita et al. [38] found that older hens (78–80 weeks) laid eggs with a significantly improved AI, TI and h/H ratio compared to younger hens (28–30 weeks). Accordingly, Ko et al. [48] reported better fat quality in terms of AI, TI and h/H ratio in eggs from medium (42 weeks)- and older (74 weeks)-aged hens than in eggs from younger hens (24 weeks). However, in a study by Peng et al. [77], which was carried out on older hens, the AI and TI were not affected by increasing layer age from 68 to 74 weeks. In contrast to the present findings, our previous report [34] showed that the addition of 3–6% OP to the diet of hens housed in enriched cages improved health lipid indices in a dose-dependent manner, with better results obtained with higher incorporation rates (5% and 6%). The different housing systems used in the two trials (cage vs. floor) may have played a role in the different results obtained, as it has been previously documented that health lipid indices are influenced by the type of production system used [77,96]. According to Laudadio et al. [61], the AI of the yolk of eggs produced by laying hens fed with polyphenol-rich extra-virgin olive oil decreased linearly with increasing levels of dietary polyphenols. Previous similar studies conducted on other farm animals have demonstrated that incorporating OP into their diets enhances the health lipid indices of final products, such as ewe milk [97] and cheese [29], dairy cheese [30], rabbit meat [24] and beef meat [31], by lowering both the AI and TI and increasing the h/H ratio.

According to egg composition tables from different EU countries and the USA, the total fat content of eggs ranges from 8.7 to 11.2 per 100 g of whole egg [7]. In the present study, the % fat content of eggs in all the dietary treatments was within the reported reference values, ranging from 8.55% to 9.77%. It has previously been documented that, unlike the FA composition of eggs, which is strongly linked to the hen’s diet, the total lipid content is relatively stable and very difficult to change [7]. In line with this documentation, we found that the % egg fat content was only affected by hen age and not by the supplementation of OP in the diet of laying hens. Consistent with the findings of many researchers, the eggs from older hens in this study had higher concentrations of % total fat than the eggs from younger birds [75,76,79,81,89,98]. However, Ko et al. [48] found that egg yolks from different age groups (early: 24 weeks; intermediate: 42 weeks; and late: 74 weeks) had similar crude lipid percentages. The higher fat content seen in eggs from older hens indicates that they deposit more yolk and fat in their eggs, which is associated with changes in yolk size [81]. The existing data on the dietary effect of OP in total egg lipids are rather inconsistent. In a previous report, we found that the fat content of the eggs increased at incorporation rates from 3% to 5% OP, but at the dietary level of 6%, this parameter was significantly reduced compared to the CON eggs [34]. A similar (decreasing) dietary effect of OP on egg fat concentration was also observed by other researchers in similar feeding trials with laying hens [33] and Japanese laying quails [99].

Differences in OP composition, diet formulation, inclusion rates, hen age and hybrids may explain the variability in our results regarding the dietary effects of OP on the egg nutritional traits evaluated in the present study compared to results from similar feeding trials. On the other hand, the age-related differences observed in the similar studies mentioned may be linked to variations in hen metabolism, which depend on the animal’s age and affect changes in the yolk’s FA composition [100]. Additionally, the impact of hen age is known to be influenced by factors such as breed, diet and feed composition, as well as environmental conditions [81]. The differences in the housing system used in this study may also have led to the differences in the results obtained, as this has been shown to affect the FA composition of eggs [77].

## 5. Conclusions

The present study showed that the nutritional value of eggs was influenced by both OP supplementation and the age of the hens. According to our findings, the addition of OP to the hens’ diet reduced the cholesterol content of the eggs compared to the control group, with particularly noticeable effects observed during the second phase of the laying period. Furthermore, during the first phase of the laying period, the OP6 eggs were found to have a higher concentration of total phenolics than the controls and a higher amount of PUFAs than the other two groups. As the birds aged, the total phenolic content, cholesterol, n3 PUFAs and % fat content of the eggs increased in all the groups, while the concentrations of triglycerides and oleuropein decreased. Regarding the FA profile, the concentration of MUFAs was significantly higher in the eggs of the birds fed with OP than in the controls during the second phase of the laying period. However, as the birds aged, the concentration of SFAs in the OP eggs decreased, although no significant differences between the groups were observed during this period. With the increasing age of the hens, the nutritional value of the eggs improved for all the dietary treatments, as indicated by the lower n6/n3 PUFA ratio, the lower AI (recorded only in the OP eggs) and TI, and the higher h/H ratio. Overall, our results showed that the inclusion of OP in the diets of laying hens at both levels and at both stages of the production cycle studied had a beneficial effect on the nutritional value of eggs, providing potential health benefits for consumers. The observed improvement in health lipid indices with hen age highlighted eggs from older hens as a valuable source of high-quality fats. Based on these results, it is highly recommended to include OP in the diet of laying hens in alternative housing systems at levels of 4% and 6% to improve the nutritional value of their eggs.

## Figures and Tables

**Table 1 foods-13-04152-t001:** Concentration of saturated (SFAs), monounsaturated (MUFAs) and polyunsaturated (PUFAs) fatty acids (g/100 g fat), total phenolics, cholesterol, triglycerides, oleuropein and lipid health indices assessed in the whole edible part of eggs collected from 39- and 59-week-old hens from all groups. Group, age and group × age effects are shown. Data are presented as marginal mean ± SE.

Parameter	Group	WK39	WK59	*Group Mean*	*p*		
					Group	Age	Group × Age
Total Phenolics (ppm)	CON	55.14 ± 2.18 ^aA^	96.87 ± 2.29 ^B^	*76.00 ± 1.58*	0.120	<0.001	0.004
	OP4	60.32 ± 2.18 ^abA^	100.31 ± 2.29 ^B^	*80.31 ± 1.58*			
	OP6	66.11 ± 2.18 ^bA^	93.47 ± 2.29 ^B^	*79.79 ± 1.58*			
	*Age mean*	*60.52 ± 1.26 ^A^*	*96.88 ± 1.32 ^B^*				
Cholesterol (ppm)	CON	2487.00 ± 84.56 ^A^	3414.44 ± 89.13 ^aB^	*2950.72* *± 61.43 ^a^*	0.005	<0.001	0.537
	OP4	2288.00 ± 84.56 ^A^	3063.33 ± 89.13 ^bB^	*2675.68 ± 61.43 ^b^*			
	OP6	2231.00 ± 84.56 ^A^	3187.78 ± 89.13 ^bB^	*2709.39 ± 61.43 ^b^*			
	*Age mean*	*2335.33 ± 48.82 ^A^*	*3221.85 ± 51.46 ^B^*				
Triglycerides (%)	CON	8.45 ± 0.12	8.00 ± 0.12	*8.23 ± 0.09*	0.597	0.022	0.226
	OP4	8.12 ± 0.12	8.09 ± 0.12	*8.10 ± 0.09*			
	OP6	8.29 ± 0.12	8.08 ± 0.12	*8.18 ± 0.09*			
	*Age mean*	*8.29 ± 0.07 ^A^*	*8.06 ± 0.07 ^B^*				
Oleuropein (ppm)	CON	20.96 ± 0.42	19.90 ± 0.45	*20.43 ± 0.31*	0.948	0.002	0.956
	OP4	21.10 ± 0.42	20.04 ± 0.45	*20.57 ± 0.31*			
	OP6	21.15 ± 0.42	19.87 ± 0.45	*20.51 ± 0.31*			
	*Age mean*	*21.07 ± 0.25 ^A^*	*19.94 ± 0.26 ^B^*				
SFAs (g100 g Fat)	CON	32.24 ± 0.28	31.58 ± 0.29	*31.91 ± 0.20*	0.578	<0.001	0.145
	OP4	32.84 ± 0.28 ^A^	31.36 ± 0.29 ^B^	*32.10 ± 0.20*			
	OP6	33.07 ± 0.28 ^A^	31.32 ± 0.29 ^B^	*32.20 ± 0.20*			
	*Age mean*	*32.72 ± 0.16 ^A^*	*31.42 ± 0.17 ^B^*				
MUFAs (g/100 g Fat)	CON	56.71 ± 0.28 ^a^	55.81 ± 0.29 ^a^	*56.26 ± 0.20 ^a^*	<0.001	<0.001	<0.001
	OP4	56.35 ± 0.28 ^aA^	57.68 ± 0.29 ^bB^	*57.01 ± 0.20 ^b^*			
	OP6	54.06 ± 0.28 ^bA^	57.45 ± 0.29 ^bB^	*55.75 ± 0.20 ^a^*			
	*Age mean*	*55.71 ± 0.16 ^A^*	*56.98 ± 0.17 ^B^*				
PUFAs (gr100 g Fat)	CON	11.05 ± 0.34 ^aA^	12.61 ± 0.35 ^aB^	*11.83 ± 0.24 ^a^*	0.003	0.933	<0.001
	OP4	10.81 ± 0.34 ^a^	10.96 ± 0.35 ^b^	*10.89 ± 0.24 ^b^*			
	OP6	12.86 ± 0.34 ^bA^	11.23 ± 0.35 ^abB^	*12.05 ± 0.24 ^a^*			
	*Age mean*	*11.58 ± 0.19*	*11.60 ± 0.20*				
PUFAs/SFAs	CON	0.34 ± 0.01 ^abA^	0.40 ± 0.01 ^B^	*0.37 ± 0.01 ^a^*	0.018	0.147	0.004
	OP4	0.33 ± 0.01 ^a^	0.35 ± 0.01	*0.34 ± 0.01 ^b^*			
	OP6	0.39 ± 0.01 ^b^	0.36 ± 0.01	*0.38 ± 0.01 ^a^*			
	*Age mean*	*0.35 ± 0.01*	*0.37 ± 0.01*				
n6 PUFAs	CON	10.57 ± 0.31 ^a^	11.63 ± 0.33 ^a^	*11.10 ± 0.23 ^a^*	0.002	0.215	<0.001
	OP4	10.27 ± 0.31 ^a^	10.13 ± 0.33 ^b^	*10.20 ± 0.23 ^b^*			
	OP6	12.31 ± 0.31 ^bA^	10.41 ± 0.33 ^abB^	*11.36 ± 0.23 ^a^*			
	*Age mean*	*11.05 ± 0.18*	*10.72 ± 0.19*				
n3 PUFAs	CON	0.43 ± 0.02 ^A^	0.85 ± 0.03 ^aB^	*0.64 ± 0.02*	0.100	<0.001	0.001
	OP4	0.46 ± 0.02 ^A^	0.72 ± 0.03 ^bB^	*0.59 ± 0.02*			
	OP6	0.47 ± 0.02 ^A^	0.72 ± 0.03 ^bB^	*0.60 ± 0.02*			
	*Age mean*	*0.45 ± 0.01 ^A^*	*0.77 ± 0.01 ^B^*				
n6 PUFAs/n3 PUFAs	CON	24.97 ± 0.59 ^abA^	13.65 ± 0.62 ^B^	*19.31 ± 0.43 ^ab^*	0.003	<0.001	0.003
	OP4	22.52 ± 0.59 ^aA^	14.27 ± 0.62 ^B^	*18.40 ± 0.43 ^a^*			
	OP6	26.77 ± 0.59 ^bA^	14.37 ± 0.62 ^B^	*20.57 ± 0.43 ^b^*			
	*Age mean*	*24.75 ± 0.34 ^A^*	*14.09 ± 0.36 ^B^*				
AI	CON	0.49 ± 0.006	0.48 ± 0.007	*0.49 ± 0.005*	0.590	<0.001	0.135
	OP4	0.51 ± 0.006 ^A^	0.47 ± 0.007 ^B^	*0.49 ± 0.005*			
	OP6	0.51 ± 0.006 ^A^	0.47 ± 0.007 ^B^	*0.49 ± 0.005*			
	*Age mean*	*0.50 ± 0.004 ^A^*	*0.47 ± 0.004 ^B^*				
TI	CON	0.92 ± 0.01 ^A^	0.86 ± 0.01 ^B^	*0.89 ± 0.01*	0.445	<0.001	0.375
	OP4	0.94 ± 0.01 ^A^	0.86 ± 0.01 ^B^	*0.90 ± 0.01*			
	OP6	0.95 ± 0.01 ^A^	0.86 ± 0.01 ^B^	*0.91 ± 0.01*			
	*Age mean*	*0.94 ± 0.01 ^A^*	*0.86 ± 0.01 ^B^*				
h/H	CON	2.35 ± 0.03 ^A^	2.50 ± 0.03 ^B^	*2.43 ± 0.02*	0.257	<0.001	0.727
	OP4	2.27 ± 0.03 ^A^	2.47 ± 0.03 ^B^	*2.37 ± 0.02*			
	OP6	2.30 ± 0.03 ^A^	2.49 ± 0.03 ^B^	*2.39 ± 0.02*			
	*Age mean*	*2.30 ± 0.02 ^A^*	*2.49 ± 0.02 ^B^*				

^a,b^ Means within columns differ significantly (*p* < 0.05). ^A,B^ Means within rows differ significantly (*p* < 0.05). AI: atherogenic index; TI: thrombogenic index; h/H: hypocholesterolemic-to-hypercholesterolemic FA ratio.

**Table 2 foods-13-04152-t002:** Total fat content % and fatty acid composition (g/100 g fat) as evaluated in whole edible parts of eggs collected from 39- and 59-week-old hens of all groups. Group, age, and group × age effects are shown. Data are presented as marginal mean ± SE.

Parameter	Group	WK39	WK59	*Group Mean*	*p*		
					Group	Age	Group × Age
Fat %	CON	8.80 ± 0.18 ^A^	9.57 ± 0.19 ^B^	*9.18 ± 0.13*	0.897	<0.001	0.264
	OP4	8.50 ± 0.18 ^A^	9.77 ± 0.19 ^B^	*9.14 ± 0.13*			
	OP6	8.86 ± 0.18 ^A^	9.59 ± 0.19 ^B^	*9.22 ± 0.13*			
	*Age mean*	*8.72 ± 0.10 ^A^*	*9.64 ± 0.11 ^B^*				
Fatty acids (gr/100 g fat)							
Myristic acid (C14:0)	CON	0.390 ± 0.007	0.411 ± 0.007	*0.401 ± 0.005*	0.382	0.528	0.063
	OP4	0.407 ± 0.007	0.394 ± 0.007	*0.401 ± 0.005*			
	OP6	0.391 ± 0.007	0.393 ± 0.007	*0.392 ± 0.005*			
	*Age mean*	*0.396 ± 0.004*	*0.400 ± 0.004*				
Myristoleic acid (C14:1)	CON	0.095 ± 0.004 ^ab^	0.089 ± 0.004	*0.092 ± 0.003 ^ab^*	0.012	0.084	0.076
	OP4	0.107 ± 0.004 ^a^	0.091 ± 0.004	*0.099 ± 0.003 ^a^*			
	OP6	0.084 ± 0.004 ^b^	0.088 ± 0.004	*0.086 ± 0.003 ^b^*			
	*Age mean*	*0.095 ± 0.002*	*0.089 ± 0.003*				
Palmitic acid (C16:0)	CON	26.172 ± 0.229 ^A^	25.078 ± 0.242 ^B^	*25.625 ± 0.167*	0.437	<0.001	0.650
	OP4	26.687 ± 0.229 ^A^	25.163 ± 0.242 ^B^	*25.925 ± 0.167*			
	OP6	26.518 ± 0.229 ^A^	25.130 ± 0.242 ^B^	*25.824 ± 0.167*			
	*Age mean*	*26.459 ± 0.132 ^A^*	*25.124 ± 0.140 ^B^*				
Palmitoleic acid (C16:1)	CON	4.604 ± 0.099 ^abA^	3.740 ± 0.105 ^aB^	*4.172 ± 0.072 ^a^*	<0.001	0.003	<0.001
	OP4	4.841 ± 0.099 ^b^	4.781 ± 0.105 ^b^	*4.811 ± 0.072 ^b^*			
	OP6	4.257 ± 0.099 ^a^	4.397 ± 0.105 ^b^	*4.327 ± 0.072 ^a^*			
	*Age mean*	*4.567 ± 0.057 ^A^*	*4.306 ± 0.061 ^B^*				
Heptadecanoic acid (C17:0)	CON	0.084 ± 0.004 ^aA^	0.149 ± 0.004 ^aB^	*0.116 ± 0.003 ^a^*	<0.001	<0.001	<0.001
	OP4	0.077 ± 0.004 ^aA^	0.119 ± 0.004 ^bB^	*0.098 ± 0.003 ^b^*			
	OP6	0.102 ± 0.004 ^bA^	0.130 ± 0.004 ^bB^	*0.116 ± 0.003 ^a^*			
	*Age mean*	*0.088 ± 0.002 ^A^*	*0.133 ± 0.002 ^B^*				
Cis-10-Heptadecenoic acid (C17:1)	CON	ND*	0.144 ± 0.003 ^a^	*0.072 ± 0.002 ^a^*	<0.001	<0.001	<0.001
	OP4	ND	0.119 ± 0.003 ^b^	*0.059 ± 0.002 ^b^*			
	OP6	ND	0.130 ± 0.003 ^b^	*0.065 ± 0.002 ^b^*			
	*Age mean*	ND	*0.131 ± 0.002*				
Stearic acid (C18:0)	CON	5.550 ± 0.093 ^a^	5.879 ± 0.099	*5.714 ± 0.068*	0.169	0.640	0.001
	OP4	5.631 ± 0.093 ^a^	5.620 ± 0.099	*5.625 ± 0.068*			
	OP6	6.024 ± 0.093 ^bA^	5.596 ± 0.099 ^B^	*5.810 ± 0.068*			
	*Age mean*	*5.735 ± 0.054*	*5.698 ± 0.057*				
Elaidic acid (C18:1n9t)	CON	0.079 ± 0.003 ^A^	0.061 ± 0.003 ^B^	*0.070 ± 0.002*	0.568	<0.001	0.224
	OP4	0.076 ± 0.003 ^A^	0.058 ± 0.003 ^B^	*0.067 ± 0.002*			
	OP6	0.083 ± 0.003 ^A^	0.056 ± 0.003 ^B^	*0.069 ± 0.002*			
	*Age mean*	*0.079 ± 0.002 ^A^*	*0.058 ± 0.002 ^B^*				
Oleic acid (C18:1n9c)	CON	51.602 ± 0.282 ^a^	51.678 ± 0.297	*51.640 ± 0.205 ^ab^*	0.018	<0.001	<0.001
	OP4	50.992 ± 0.282 ^aA^	52.551 ± 0.297 ^B^	*51.772 ± 0.205 ^a^*			
	OP6	49.262 ± 0.282 ^bA^	52.681 ± 0.297 ^B^	*50.972 ± 0.205 ^b^*			
	*Age mean*	*50.619 ± 0.163 ^A^*	*52.303 ± 0.172 ^B^*				
Linoleic acid (C18:2n6c)	CON	10.449 ± 0.309 ^a^	11.479 ± 0.326 ^a^	*10.964 ± 0.225 ^a^*	0.002	0.195	<0.001
	OP4	10.145 ± 0.309 ^a^	9.986 ± 0.326 ^b^	*10.065 ± 0.225 ^b^*			
	OP6	12.170 ± 0.309 ^bA^	10.279 ± 0.326 ^abB^	*11.224 ± 0.225 ^a^*			
	*Age mean*	*10.921 ± 0.178*	*10.581 ± 0.188*				
γ-Linolenic acid (C18:3n6)	CON	0.060 ± 0.004 ^A^	0.082 ± 0.004 ^B^	*0.071 ± 0.003*	0.796	0.007	0.016
	OP4	0.069 ± 0.004	0.074 ± 0.004	*0.072 ± 0.003*			
	OP6	0.074 ± 0.004	0.073 ± 0.004	*0.074 ± 0.003*			
	*Age mean*	0.068 ± 0.002 ^A^	0.077 ± 0.002 ^B^				
Linolenic acid (C18:3n3)	CON	0.133 ± 0.010 ^A^	0.462 ± 0.010 ^aB^	*0.298 ± 0.007 ^a^*	0.009	<0.001	<0.001
	OP4	0.146 ± 0.010 ^A^	0.389 ± 0.010 ^bB^	*0.267 ± 0.007 ^b^*			
	OP6	0.137 ± 0.010 ^A^	0.406 ± 0.010 ^bB^	*0.271 ± 0.007 ^b^*			
	*Age mean*	*0.139 ± 0.006 ^A^*	*0.419 ± 0.006 ^B^*				
Cis-11-Eicosenoic (C20:1)	CON	0.332 ± 0.009 ^aA^	0.122 ± 0.010 ^B^	*0.227 ± 0.007 ^ab^*	0.015	<0.001	0.013
	OP4	0.332 ± 0.009 ^aA^	0.090 ± 0.010 ^B^	*0.211 ± 0.007 ^a^*			
	OP6	0.375 ± 0.009 ^bA^	0.106 ± 0.010 ^B^	*0.240 ± 0.007 ^b^*			
	*Age mean*	*0.346 ± 0.005 ^A^*	*0.106 ± 0.006 ^B^*				
Cis-11,14-Eicosadienoic acid (C20:2)	CON	0.053 ± 0.005 ^a^	0.072 ± 0.005	*0.063 ± 0.004 ^a^*	0.012	0.211	<0.001
	OP4	0.073 ± 0.005 ^ab^	0.063 ± 0.005	*0.068 ± 0.004 ^ab^*			
	OP6	0.091 ± 0.005 ^bA^	0.066 ± 0.005 ^B^	*0.078 ± 0.004 ^b^*			
	*Age mean*	*0.072 ± 0.003*	*0.067 ± 0.003*				
Cis-8,11,14-Eicosatrienoic (C20:3n6)	CON	0.063 ± 0.005	0.070 ± 0.006	*0.066 ± 0.004*	0.804	0.683	0.404
	OP4	0.060 ± 0.005	0.066 ± 0.006	*0.063 ± 0.004*			
	OP6	0.068 ± 0.005	0.061 ± 0.006	*0.065 ± 0.004*			
	*Age mean*	*0.064 ± 0.003*	*0.066 ± 0.003*				
Cis-11,14,17-Eicosatrienoate acid (C20:3n3)	CON	0.293 ± 0.016 ^A^	0.389 ± 0.017 ^aB^	*0.341 ± 0.011*	0.455	0.009	0.006
	OP4	0.312 ± 0.016	0.333 ± 0.017 ^ab^	*0.323 ± 0.011*			
	OP6	0.329 ± 0.016	0.319 ± 0.017 ^b^	*0.324 ± 0.011*			
	*Age mean*	0.311 ± 0.009 ^A^	0.347 ± 0.010 ^B^				

^a,b^ Means within columns differ significantly (*p* < 0.05). ^A,B^ Means within rows differ significantly (*p* < 0.05). ND: not detected.

## Data Availability

The data presented in this study are available on request from the corresponding author (Dr. Anna Dedousi).

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
