# Peer review of "The Beneficial Dietary Effect of Dried Olive Pulp on Some Nutritional Characteristics of Eggs Produced by Mid- and Late-Laying Hens"

_foods, 2024, doi:10.3390/foods13244152_

Round 1
Reviewer 1 Report
Comments and Suggestions for Authors
The research was presented well however, it needs some modifications, including standardizing the writing of a significant level throughout the manuscript.
- The abstract must be reformulated to include the problem, objects, methods, and most important results.
- It must be clarified at what age the hens were fed OP and how long they were fed OP and took egg samples. Is there a pre-trial period?
Reviewer 2 Report
Comments and Suggestions for Authors
1. Figures can be appropriately cited to reflect the problem, which can be more engaging and enhance the reader's understanding.
2. There is a large discussion of the influence of chicken age on egg quality in the conclusion and discussion, but it is not highlighted in the title and introduction, and is only mentioned in passing.
3. The logic of the whole text is confusing, and it should be discussed gradually around the core point of the article.
4. The amount of data is small, while the data form is single. There are too many words, but the organization is unclear.
5. Some of the literature cited is not very relevant to the content of the study.
Reviewer 3 Report
Comments and Suggestions for Authors
Author Response
Please see the attachment

Reviewer 4 Report
Comments and Suggestions for Authors
Dear authors, thanks for this submission in Foods J. The concept of this manuscript is fine. Authors explained well to focus the background of uses olive pulp in egg quality. Please check my below comments...
Introduction part : well described
L5-51: rewrite the statement
2.2: experimental design: Please provide the replication number in text, and how many birds were considered per replication, also flock size.
2.3: sample eggs were collected on weekly basis? Mention it clearly in text. If so, then how many eggs were selected weekly from each replication?
2.3.1: which part of egg was considered, either only yolk or albumin or mixed. Mention clearly the part of egg sample in text for section 2.3.2 and 2.3.3, 2.3.4 and 2.3.5
How and what amount of sample was considered, mention in brief sothat the future researchers can follow your protocols.
2.3.5: for which sample?
Result: 3.1: Please provide all information regarding the table 3.1 and place table just after text information.
Where is 3.2?
Make a sub heading for information table 2. From L 319.
Table 1: From the mean value of Cholesterol- delete the fraction, also delete fraction from SE value. (example: CON 2487 ±84), make similar for all means.
Consider for table 2 also.
Discussion : well discussed, can minimize the content.
Conclusion: A final statement regarding the uses of OP with recommended dosages is needed at end of conclusion.
Round 2
Reviewer 3 Report
Comments and Suggestions for Authors
The authors have provided a reasonable explanation for my concerns, and I believe it can be accepted now. Good luck!